# Overpumping leads to California groundwater arsenic threat

Ryan Smith [1], Rosemary Knight[1] & Scott Fendorf [2]

Water resources are being challenged to meet domestic, agricultural, and industrial needs. To complement finite surface water supplies that are being stressed by changes in precipitation and increased demand, groundwater is increasingly being used. Sustaining groundwater use requires considering both water quantity and quality. A unique challenge for groundwater use, as compared with surface water, is the presence of naturally occurring contaminants within aquifer sediments, which can enter the water supply. Here we find that recent groundwater pumping, observed through land subsidence, results in an increase in aquifer arsenic concentrations in the San Joaquin Valley of California. By comparison, historic groundwater pumping shows no link to current groundwater arsenic concentrations. Our results support the premise that arsenic can reside within pore water of clay strata within aquifers and is released due to overpumping. We provide a quantitative model for using subsidence as an indicator of arsenic concentrations correlated with groundwater pumping.

[1] Department of Geophysics, Stanford University, 397 Panama Mall, Stanford, California 94305, USA. [2] Department of Earth System Science, Stanford University, 473 Via Ortega, Stanford, California 94305, USA. Correspondence and requests for materials should be addressed to R.S. (email: rgsmith@stanford.edu)

Globally, groundwater provides almost half of all drinking water[1], making it one of the world's most important resources. Within the United States, the Central Valley of California accounts for roughly 20% of groundwater withdrawals[2]. The Central Valley is an arid region that supports a $17 billion agricultural industry. In the southern, highly productive region of the valley (known as the San Joaquin Valley), aquifers are particularly stressed due to the high water demands. Further, groundwater is the main source of drinking water for roughly one million people in the San Joaquin Valley. Herein, we focus on threats to groundwater quality induced by naturally occurring arsenic, which may have devastating impacts on both human health and food production.

Arsenic is a ubiquitous, naturally occurring contaminant that is a common problem in many aquifers that are pumped for drinking water[3]; most notably, it is presently having a devasting impact on groundwater quality throughout Asia[4]. When present in significant amounts, it increases the risk of cancer, heart disease, and diabetes[5]. Hazardous levels of arsenic typically result from anaerobic conditions, as noted for the shallow, Holocene aquifers of Asia[4], or from high pH (pH > 8.5) often observed in high arsenic regions of the Andes[6]. In addition, overpumping an aquifer system has been noted to increase the arsenic levels in Southeast Asia by drawing arsenic from less-permeable anaerobic clay strata into the aquifer[7].

Arsenic within pumped groundwater of the San Joaquin Valley has been noted for decades. Approximately 10% of the wells tested within the last 10 years have shown arsenic (As) concentrations above 10 µg/L (p.p.b.), the level recommended as the maximum acceptable by the World Health Organization (WHO). As groundwater is increasingly being pumped to meet agricultural and domestic needs, preserving groundwater resources (i.e., maintaining water quality) is imperative. Here we seek to determine the source of arsenic contamination within groundwater of the Central Valley, CA, and to develop quantitative means for predicting degradation of groundwater quality. We focus our study on the Tulare basin (Fig. 1), a highly productive agricultural region of the San Joaquin Valley where groundwater is essential for meeting water demands.

Groundwater pumping in the San Joaquin Valley has caused declines of ~ 60 m in groundwater levels over the past century, leading to subsidence as high as ~ 9 m from 1925 to 1970, a rate of 20 cm/year[8]. Rates of subsidence as high as 25 cm/year have been observed in more recent droughts spanning from 2007 to 2010 and from 2012 to 2016[9, 10]. Subsidence due to groundwater pumping is caused by a pore pressure drop in aquifer materials, which increases the effective stress, $\sigma_e$, defined as $\sigma_e = \sigma_T - P_P$, where $\sigma_T$ is the total stress and $P_P$ is the pore pressure[11]. Increasing the effective stress results in aquifer compaction. The majority of compaction and linked subsidence of the overlying ground surface is caused by drainage of clays due to their weaker geomechanical properties, as indicated by their higher skeletal specific storage. Thus, the high subsidence levels provide a measure of overpumping in aquifers having substantial clay content, such as the aquifers of the San Joaquin Valley. In this region, the upper 500 m, which is the greatest depth typically drilled for groundwater pumping, consists of alternating layers of sand, gravel, and clay. In general, the aquifer system is divided into an upper aquifer, a thick clay confining unit known as the Corcoran clay, and a lower aquifer. The upper and lower aquifers contain sands and gravels, as well as numerous thin clay layers[12]. As a measure of overpumping, we use subsidence, derived from Interferometric Synthetic Aperture Radar (InSAR) data, to predict arsenic levels quantitatively.

We find that subsidence data have a strong correlation with concurrent arsenic concentrations. This demonstrates a quantitative link between overpumping of groundwater systems and arsenic contamination, a mechanism that has been proposed previously[7] but never statistically demonstrated.

## Results and Discussion

**Modeling arsenic concentration**. We integrated estimates of subsidence with additional variables known from previous studies[13] to affect arsenic levels into a random forest model that accounts for nonlinear relationships, as well as interdependencies of different variables. Two models were developed, one that predicts recent (2007 to 2015) arsenic concentrations and a second that predicts historic (1986 to 1993) arsenic concentrations. These dates were chosen because they both span long droughts in the San Joaquin Valley, during which time appreciable groundwater decline and subsidence occurred (Supplementary Fig. 5). We did not include the mild-drought conditions from 1999 to 2005, because we had limited subsidence data over that period, and because we considered more intense droughts to produce a stronger signal for our analysis. We used the output of the random forest models to assess the effect of overpumping as indicated by subsidence on arsenic levels. We found that in both models, concurrent subsidence markedly increases the risk of arsenic contamination. As both models establish similar relationships (capturing the time periods of recent and past periods of drought), we focus our discussion on the recent arsenic concentration model, pointing out where key differences exist. The full results of both models are shown in the Supplementary Information (Supplementary Figs. 1-4)

**Mechanism for arsenic contamination from overpumping**. The relationship between subsidence and arsenic concentration is linked to clay strata (Fig. 1). It has been noted that clay particles become enriched in arsenic due to their reactivity and high surface area to volume ratio relative to sand-sized particles[4]. Arsenic has been transported to the San Joaquin Valley from the Sierra Nevada and coastal mountain ranges by rivers, which cut through arsenic-bearing formations, for millions of years[14]. Clays at or near the surface at the time of deposition are the primary host of transported arsenic, which has been shown to adsorb on clay surfaces in significant amounts in the San Joaquin Valley[15], in a process similar to what occurs in other sedimentary basins, including those throughout Asia[4]. As the clays are buried over geologic time, their restricted oxygen supply results in reduction of the arsenic at depths > 60 m[14], resulting in dissolution of the arsenic within the clay pore water[7]. This mechanism for arsenic concentration in clays is supported by other studies in the area[13], who have found a positive relationship between aquifer clay content and arsenic concentration in the San Joaquin Valley.

When unperturbed, groundwater within the aquifer primarily flows horizontally through the sediments with highest permeability, typically sands and gravels. Thus, pumped groundwater comes mostly from sands and gravels, which have lower arsenic concentrations. When the aquifer system is stressed from overpumping, high vertical hydraulic gradients cause a larger volume of water to be drawn from less-permeable clays, inducing the release of water with high arsenic concentrations. As a consequence, overpumping increases arsenic concentrations, as noted for the lower Mekong Delta, Vietnam[7]. Concomitant with decreased pore pressures is compaction of the aquifer and resulting land subsidence. Thus, the dual consequence of overpumping is land subsidence and increased extraction of pore water from clay layers, giving rise to a link between land subsidence and groundwater arsenic concentrations.

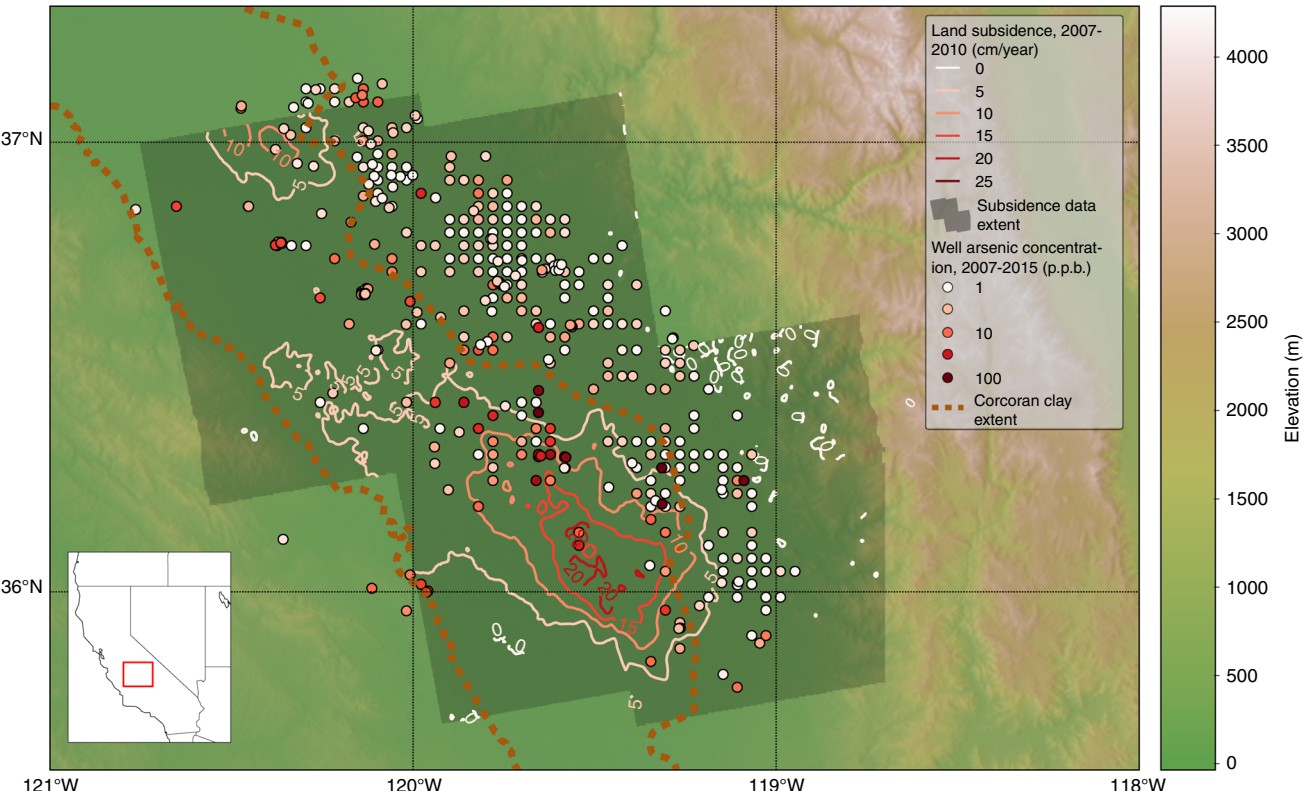

**Fig. 1** Comparison of subsidence, arsenic concentration, and clay extent within the San Joaquin Valley, CA. Arsenic concentrations increase toward the center of the valley where subsidence is more extensive and a confining clay layer (known as the Corcoran clay) is present[12]. The grayed-out region is the area where recent subsidence data (obtained from InSAR) were processed. Arsenic concentrations vary by orders of magnitude and are thus shown with a logarithmic color bar (see Supplementary Fig. 8 for arsenic concentration histograms). The arsenic concentration data points were sourced from[17]. The basemap was created using elevation data from the Shuttle Radar Topography Mission[18]

**Quantifying the impact of overpumping on groundwater arsenic concentrations**. We highlighted four of the most important variables describing arsenic concentration within the Tulare Basin in the recent model, shown in Fig. 2a-d. Of these, the thickness of the Corcoran Clay (a confining unit that overlies a lower aquifer) shows a positive correlation with arsenic concentrations due to increased clay content. Elevation has a negative correlation, as lower areas are more likely to have been water-saturated and thus anaerobic. A positive correlation was found between $\log_{10}(Mn)$ and arsenic concentrations, as the presence of manganese indicates an anoxic environment, in which arsenic tends to be more soluble. Significantly, recent subsidence from InSAR showed a positive correlation, as overpumping leads to increased pore water drainage from clays. The first three variables are well-known from the literature and not related to human activity. The quantitative link between pumping-induced subsidence and arsenic concentrations has not been shown before, and is directly related to human activity.

By comparing the historic and recent arsenic concentration models, we see the transient nature of the relationship between pumping-induced subsidence and arsenic concentrations (Fig. 2e). Although historic subsidence has a high impact on historic arsenic concentrations, it has virtually no impact on recent arsenic concentrations. A lack of correlation between present arsenic concentrations and historic subsidence (related to overpumping) suggests that arsenic levels due to overpumping slowly return to their original levels after the groundwater pumping is decreased, implying that arsenic is flushed from the aquifer. Thus, our findings indicated that avoiding overpumping of aquifers,

easily seen in InSAR data, should improve water quality, at least for the San Joaquin Valley.

The historic arsenic concentration model shows a weaker, but still positive, correlation between Corcoran clay thickness and arsenic concentration, as well as manganese concentration and arsenic concentration. Due to the dynamic nature of groundwater chemistry, the relative importance of different mechanisms for arsenic contamination change over time, highlighting the need for regular calibration with up to date datasets.

To further quantify the impact of overpumping on arsenic concentrations, we simulated three different hypothetical scenarios with our predictive random forest model: oxic, suboxic, and anoxic subsurface conditions. We used a variation of the partial dependence plot that averages all variables except subsidence, which is varied from 0 to 10 cm/year, and $\log_{10}(Mn)$, which is set at 0, 1, and 2 for oxic, suboxic, and anoxic scenarios, respectively, to estimate the probability that the predicted arsenic levels exceeded the WHO's standard of 10 µg/L arsenic (Fig. 3). At ca. 8 cm/year subsidence, the probability that arsenic exceeds the WHO standard dramatically accelerates. This inflection point is also present at roughly the same location in the historic arsenic model (Fig. 2e). As inelastic deformation of clay beds begins, the rate of subsidence increases significantly[10]. As inelastic deformation occurs when pore pressure drops below the lowest level previously experienced, it releases pore water that has not been mixed with the main aquifer previously, which in the present case results in arsenic-rich pore water being captured within the pumped groundwater.

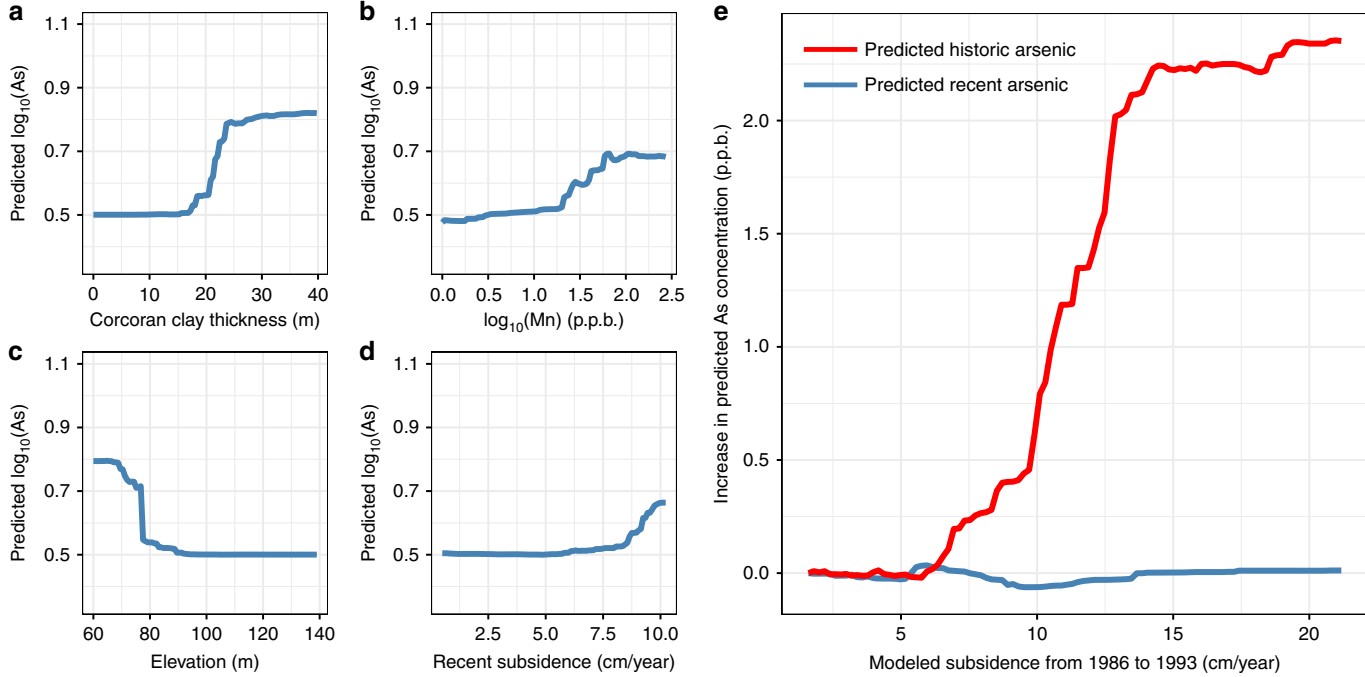

**Fig. 2** Partial dependence plots of primary descriptors of groundwater arsenic concentrations. These include **a** confining clay thickness, **b** dissolved Mn concentrations, **c** elevation, and **d** recent subsidence. **e** Comparison of partial dependence plots of historic and recent subsidence. Note that historic subsidence has a great effect on historic arsenic levels but little effect on recent arsenic levels (**e**), whereas recent subsidence has a great effect on recent arsenic levels (**d**). All blue lines are derived from models predicting recent (2007–2015) arsenic concentrations, whereas the red line is derived from models predicting historic (1986–1993) arsenic concentrations

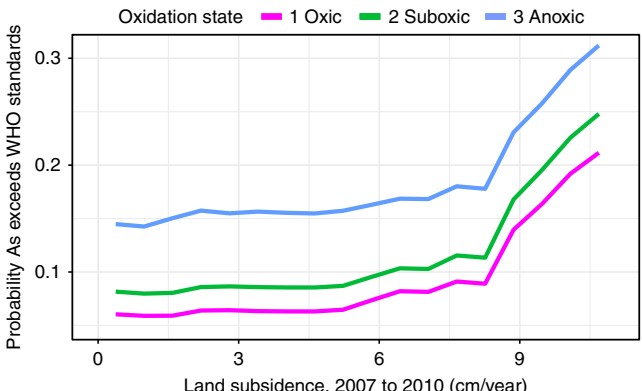

**Fig. 3** Probability that arsenic levels exceeded the WHO standard. This was calculated based on recent subsidence for each arsenic risk category

Land subsidence due to overpumping increases the probability that groundwater is contaminated beyond the WHO drinking water standard by a factor of 2 to 3 for the San Joaquin Valley. Importantly, decreasing pumping below the threshold of inestatic aquifer compression will decrease arsenic concentrations and the aquifers can recover to normal levels if overpumping is halted. Thus, subsidence maps produced by InSAR provide a means of measuring the increased risk of arsenic contamination due to overpumping of aquifers, a critically important factor as groundwater is the main source of drinking water for roughly one million people in our study area alone[16]. Moreover, with a global trend toward increase use of groundwater, effectively managing water quality with quantity is essential to preserve the use of this critical resource.

## Methods

**Data acquisition and processing**. There are many factors that control natural variation in arsenic concentration. We developed a statistical model using numerous datasets to asses the role each plays as a predictor of arsenic concentrations. We obtained arsenic level data, as well as proxy information on redox potential from manganese and sulfate concentrations, from 838 wells over the 2007 to 2015 time frame, and 424 wells over the 1986 to 1993 time frame, from the GAMA database[17], which monitors groundwater quality in California (http://geotracker.waterboards.ca.gov/gama/datadownload). We used a previously developed groundwater model[2] which provided us with several useful predictors of arsenic concentrations: percent of total water use from groundwater, which is directly related to pumping; total aquifer clay content, which sets an upper limit on available arsenic from clays; top and bottom of perforated well interval, both of which indicate the depth from which pumping occurred, a soft indicator of the oxidation state; subsidence from 1962 to 1976 and subsidence from 1986 to 1993, which provide historic measures of pumping-induced clay drainage. Elevation was also used as a predictor, obtained from the Shuttle Radar Topography Mission[18]; this is an indicator of closed-basin conditions, where contaminants such as arsenic tend to be concentrated. Slope was also derived from this dataset. As slope closely tracks predevelopment head[19], we used this with the clay content to estimate historic groundwater flow, which was also included as a predictor; our assumption being that enhanced groundwater flow would flush the system over time and reduce arsenic concentrations. We used average temperature data from PRISM; temperature variations have been related to changes in arsenic in previous work[13]. We also used evapotranspiration (ET) estimates (both from 2002 to 2007 and from 2007 to 2015)[20]; this can be used as a proxy of groundwater pumping that is independent of our InSAR measurements.

We chose not to include pesticide application as a predictor of groundwater arsenic contamination. Arsenic-bearing pesticides are largely restricted to lead arsenicals that were used predominantly on cotton and specific tree fruits (largely apples) but have been banned since the 1970's within the United States. Within the aerated surface soils, arsenic exists as arsenate[15], and would have limited mobility[4]. Thus, downward migration of surface applied arsenic is unlikely to contribute to groundwater contamination. Further, our analysis shows a correlation of groundwater arsenic with recent (last 10 years) subsidence and no correlation with older periods of subsidence. Our time series therefore further removes the possibility of arsenic pesticides impacting groundwater.

The arsenic levels, manganese levels and sulfate levels were obtained from GAMA. These levels were assumed to have a lowest detectable value of 2 p.p.b. from histogram analysis, so all values less than this were set to 2 p.p.b. As measurements of arsenic, sulfate, and manganese vary greatly with high outliers,

**Table 1 Summary of scenes used for InSAR processing**

| Frame (region) | Path | No. of scenes | No. of interferograms |
|---|---|---|---|
| Tulare | 218 | 15 | 82 |
| Fresno | 218 | 17 | 71 |
| Mendota | 219 | 19 | 82 |
| Porterville | 217 | 17 | 84 |

the logarithm of each was used in the statistical model to dampen the effect of outliers.

The historic groundwater flow was calculated using hydraulic conductivity ($K$) estimates across the Central Valley from the aforementioned groundwater model[2]. We assumed that the historic groundwater gradient ($\frac{\partial h}{\partial l}$) was equal to the land surface gradient[18]. We then used Darcy's equation to estimate flow:

$$q = -K\left(\frac{\partial h}{\partial l}\right) \tag{1}$$

where $q$ is the flux per unit area.

The distance to the nearest river was computed by determining the distance from each well to all major rivers in the study area, and taking the minimum distance as the distance to the nearest river.

**InSAR processing**. InSAR provides high-quality estimates of subsidence over large regions. InSAR measures subsidence over time with centimeter- to millimeter-scale accuracy over large regions with high (10 s to 100 s of meters) spatial resolution.

We acquired 68 SAR scenes from the ALOS PALSAR sensor, covering the time period from 2007 to 2010, over four frames (see Table 1) for InSAR processing from the Alaska Satellite Facility (https://www.asf.alaska.edu/). We then processed 319 interferograms using these scenes. We used 30 looks in azimuth and 10 looks in range to reduce noise in the pixels, resulting in a pixel size of roughly ~ 250 m by 250 m. We unwrapped the interferograms using the snaphu code[21]. Next, we used the small baseline subset method[22] to estimate the long-term deformation signal. This produced maps of the mean subsidence velocity for each of the four frames, which we merged to create one map of mean subsidence velocity. Overlapping areas were averaged. We then applied a moving average filter to further smooth the results, with a window size of 15 by 15 pixels. As the long-term signal dominated the total signal, seasonal fluctuations are muted over a multi-year time period. Thus, because of the temporal resolution of our arsenic measurements, we only considered the long-term signal. The resulting subsidence map is shown in Fig. 1 of the paper.

**Random forest model**. Groundwater overpumping, as indicated by subsidence, is one of many complex mechanisms that influence groundwater arsenic con-tamianation. In order to account for additional mechanisms, as well as the interplay between various mechanisms, we employed the random forest model[23]. Random forest models can account for nonlinear relationships between multiple variables and handle outliers well . The random forest model creates ntree number of decision trees. Each decision tree makes an independent prediction of the variable of interest, in our case arsenic concentration. The best estimate is the average of the estimates from all decision trees.

Each decision tree is built using a subset of the total data. The subset is selected using random sampling without replacement, or bagging. At each split (node), the variables to be considered for that node are also randomly sampled from the list of total variables. The values of the split are chosen to maximize the variance reduction, defined as the difference between the variance before the split and the sum of the variance of the points for which the split is true, and the variance of the points for which the split is false[24].

The number of variables to be considered at each node is described by the variable *mtry*. The variables ntree and mtry are known as tuning parameters. The overall fit of the model is dependent on these tuning parameters.

In practice, increasing ntree both improves the accuracy of the results and increases the computational cost of the algorithm. Typically, increasing ntree does not significantly improve the results after roughly 100. To be conservative we chose a value of 500 for ntree. We determined the optimal value for mtry using a validation dataset. We randomly selected 75% of the dataset to calibrate the random forest model, and used the remaining 25% of the dataset as the validation to test the accuracy of the model. We chose the value for mtry that minimized the mean squared error (MSE) in the validation dataset.

We created two random forest models, one that predicted arsenic concentrations from 2007 to 2015 (recent) and one that predicted arsenic concentrations from 1986 to 1993 (historic). The response variable was $\log_{10}(As)$. The variables used in calibrating both models were identical except that the recent subsidence from InSAR was not included in the historic arsenic concentrations model. In addition, the concentrations of arsenic, manganese and sulfate used to calibrate the models were taken from each respective time window (Supplementary

Table 1). The resulting values for mtry were both 5, with an MSE of 0.05 and 0.10, for the recent and historic random forest models, respectively. We used these calibrated, tuned random forest models to predict arsenic concentrations and establish relationships between the variables used in the model and arsenic concentrations.

Random forests are one of many machine learning regression algorithms. In this study, we also developed models using neural networks, support vector machines, and boosted gradient trees. Our analysis showed that random forests provided the best fit to the observed data. However, similar relationships between subsidence and arsenic were observed with these additional machine learning methods.

**Variable importance**. With random forests, the impact of each variable in the outcome can be assessed in multiple ways. One way is to randomly shuffle (per-mute) each variable, and observe the increase in MSE between predicted and observed arsenic concentrations. The higher the MSE, the more important that variable is to the outcome. Another way to assess variable importance is to use partial dependence plots.

**Partial dependence plots**. Partial dependence plots are a common way to visualize random forest models[25]. The goal of a partial dependence plot is to show how varying one variable impacts the outcome of the prediction, while accounting for all possible values from the other variables. The partial dependence plot is calculated as shown below:

$$\tilde{f}(x) = \frac{1}{n}\sum_{i=1}^{n} f(x, x_{ic}) \tag{2}$$

where $\tilde{f}(x)$ is the partial dependence, n is the number of rows of the dataset, $x$ is the predictor variable of interest, $x_{ic}$ are the values of all other variables, and f is a function representing the random forest model The output can either be the best estimate of $\log_{10}(As)$, or the probability that arsenic concentration exceeded 10 p.p. b. For the latter, we used all trees from the random forest model. Our random forest model produced 'best guess' estimates of arsenic levels, which are the mean of the estimates from all trees. It also produced an estimate for each of the 500 trees, for each observation. This provided us with a distribution which we used to determine the probability that the predicted arsenic levels exceeded the WHO standard.

Random forest models are limited in their predictions to observed data. For this reason, in the partial dependence plots of the main article (Figs. 2 and 3), we restricted the independent variable at the 5th and 95th percentile of each predictor variable. With those limits, 90% of the available data are displayed in the partial dependence plots. This reduces the likelihood of presenting results which are unconstrained by data. In the Supplementary Information section, we presented all of the data, but displayed dashed lines indicating the 5th and 95th percentiles (Supplementary Figs. 1-4).

**Data availability**. The geological, elevation, precipitation, temperature, ET, and water quality data have been cited throughout the paper, are listed in Supple-mentary Table 1, and are publicly available. The SAR data used for processing interferograms are available from the Alaska Satellite Facility following registration. The datasets developed by the authors, i.e., the processed deformation maps, estimated historic groundwater flow, and distance from each point to the nearest river, as well as the code used to produce the results, are available upon request from the corresponding author.

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

## Acknowledgements

R.S.'s contribution was supported by a National Science Foundation Fellowship (Award Number DGE-114747). S.F.'s contribution was supported, in part, by the US Department of Energy, Office of Biological and Environmental Research, Subsurface Biogeochemistry Program (Award Number DE-SC0016544) and the SLAC SFA Project, FWP 10094). We thank Joseph Ayotte and Kenneth Belitz for providing feedback on arsenic concentration databases in California.

## Author contributions

R.S., R.K., and S.F. conceived of the study. R.S. acquired, processed, and analyzed the data. R.S. developed and interpreted the models with input from S.F. and R.K. R.S., R.K., and S.F. wrote the manuscript.

## Additional information

**Competing interests:** The authors declare no competing interests.

