## [Peer Review File · Nature Communications]

Reviewers' comments:

Reviewer #1 (Remarks to the Author):

Dear Authors,

I carefully read the paper and the supplementary file provided with it as well as some relevant papers already published on the subject. The paper is about the links between aquifer over-pumping (mapped by detecting ground deformation using InSAR) and the release of arsenic from the aquitards (/clays layers) into the aquifer. It illustrates the important relation between aquifer sustainability and groundwater quality.

The methodology applied includes the use of a Machine Learning method that explores and reveals the link between different parameters, among which are considered compaction (/ground deformation), lithological data (/clay layer thickness), and arsenic concentration in groundwater. The paper is interesting and well written.

The phenomenological link between arsenic concentration and pumping has already been studied in the following paper (which btw, is missing in the reference list):

Erban, L. E., Gorelick, S. M., Zebker, H. A., & Fendorf, S. (2013). Release of arsenic to deep groundwater in the Mekong Delta, Vietnam, linked to pumping-induced land subsidence. *Proceedings of the National Academy of Sciences*, 110(34), 13751-13756.

I believe that Machine learning is interesting and will be increasingly used to explore how several observable parameters can be linked. It has the potential to trigger interesting phenomenological investigation. However, in this case of this paper, the phenomenology has already been explored. The only remaining novelty of the paper is the use of Machine learning to better transform proxies into the relevant information (here, arsenic concentration) and extrapolates point-based observation (training sites). As such, I don't believe that there is enough novelty within the paper to justify publication in *Nature Geoscience*.

Sincerely

Reviewer #2 (Remarks to the Author):

The paper provides a quantitative machine learning model to estimate the arsenic concentrations using land subsidence in the San Joaquin Valley of California, because over pumping leads to dual consequences of land subsidence and increased extraction of pore water from arsenic-concentrated clay layers. The variables of the model have been evaluated using partial dependence plots. The methodology in the paper, if broadly applicable, will be valuable and efficient to water managements.

The paper is informative with interesting contents, but some objectives are not clearly described, in particular, need more explanations on Figures 3, 4, S1, and S3 to add credibility of this work. The paper can be considered to be published after major revisions according to the following comments.

- 1) Missing Figure 2 in the main text.
- 2) Page 2 Line 40: Give full term and the abbreviation of As at its first appearance.
- 3) Page 2 Line 41: Give abbreviation of "World Health Organization", and use "WHO" consistently through the rest of the paper.
- 4) Page 3 Line 48: Typo of "Joaquin".
- 5) Page 3 Line 60: In your statement, you chose two time frames (1986-1993 and 2007-2015)

because they both span long droughts in the San Joaquin Valley. But from figure S5, the head declined from 1999 in the recent time frame. Why you pick up the dates from 2007 rather than 1999? And is there ground subsidence during 1999-2007?

6) Page 4 Figure 1:

The intervals for As concentration are 1, 3, 10, 32, 100 (~3-fold). Why you don't use equal intervals like what you did for land subsidence? Due to high outliers? Provide the histogram and explain.

The figure 1 in page 4 is a bit different from the full-page one you put separately: one has the inset and the other not, and one is rectangle and the other has been cropped into square. Put inset, and make sure the red box in the inset is in the same shape of the full-extent image.

I can't see clear spatial correlation between land subsidence and As concentration. The highest As concentration is located around 10 cm/yr land subsidence areas. Maybe it is easier to tell with the interpolation results of As concentration and also contours like what you did for land subsidence.

The areas in shadow show the InSAR coverage? Describe it in the caption.

What is the time frame(s) for the estimates of As concentration?

7) Page 4 Line 73: Provide the geologic vertical cross-section information about the confining clay layer in the text or in the supplementary. Or add reference.

8) Page 6 Figure 6:

The land subsidence is between 0 and 25 cm/yr during 2007-2011. Why you use a narrow range of 0 to ~10 cm/yr in (d)?

Here in Figure 6 you show the partial dependence plots for primary variables used in the model predicting arsenic concentration from 2007-2015 (Figure S1). There are some differences between the results of 2007-2015 and 1986-1993 (Figure S3), such as partial dependence plots for the thickness of Corcoran clay and Mn concentration. Please explain the possible reasons.

I suppose you draw (e) according to the "sub86to93" in Figures S1 and S3, and take the one in Figure S1 as the reference? Why the x axis is 0 to 2 in the supporting figures, but 0-20 cm/yr in (e)?

9) Page 7 Line 115: Explain how you infer "recent subsidence has a great effect on recent arsenic levels" from Figure 3d.

10) Page 7 Lines 125-129: Rearrange the sentences. "As the rate of subsidence increase, inelastic deformation of clay beds also increases significantly". This sentence is not rigorous.

11) Page 8 Figure 4: Be consistent about the time period of land subsidence, 2007-2010 or 2007-2011 (shown in figure 1).

12) Page 11 Lines 195: Add sensor name and time period.

13) Page 11 Line 203: Is there seasonality in the time series? If the seasonality need to be addressed in your model?

14) Supporting material: Page 11 Figure S7: Why are there voids in the estimates?

15) Supporting material: Pages 4 and 7: What is the meaning of "sub76"? Do you mean "subsidence from 1962 to 1976" (Page 10 Line 163)?

Reviewer #3 (Remarks to the Author):

Authors present a very interesting approach which have high practical application potential. Nevertheless, some points have to be clarified and probed in order to increase and strengthen the facts and evidences in which they support their proposal.

Authors based their proposal in the unproved fact that arsenic is being expelled from clay strata

due to the subsidence process, and thus subsidence maps can be used to detect where arsenic concentrations will be have place.

To prove this, authors have to present evidences (or specific references for the study site) showing that the strata clay (or the water which is filling clay pores) have a high content of arsenic.

Authors stated in line 75 that "The relationship between subsidence and arsenic concentration is linked to clay strata". Nevertheless, figure 1 shows that some of the arsenic highest concentration locations (roughly 20 %) are located outside the limits of the clay formation. How authors explain that? Is there clay samples lab analysis to affirm that arsenic comes from clay?

Other factors can be cause of anomalous arsenic concentrations, and authors have to discuss them. For example:

The use of pesticides surely has been extensive during long time in the area, as the valley is a very productive crop area. Then pesticides infiltrations could be linked to the current arsenic concentrations.

Deep ground water has having enough time to interact with rocks (bedrock or coarse sediments), consequently it could result in natural pollution by substances as arsenic.

How authors discarded those potential arsenic sources?

Other minor observations to improve are:

Sections are not clearly delimited as a typical article (i.e. introductions, methodology, results, discussion of results, conclusions, etc.).

Lines 48-50. Those data are not very current. I suggest leave it as antecedent but include updated information and their references... I suppose that this information exists since this valley is one of the most studied in the world.

Lines 40-51. Terzaghi, 1925 deal with soil consolidation due to an overburden on top of soil stratum, and not about subsidence due to groundwater withdrawal. I suggest change this reference by one dealing with the phenomenology of subsidence due to groundwater extraction.

Lines 51-53. Authors pointed out that "The majority of compaction and linked subsidence of the overlying ground surface is caused by drainage of clays due to their weaker geomechanical properties."

The term "weaker geomechanical properties" could be confusing. I suggest that it be specified which geomechanical properties.

Line 63. Figure 1 instead of Figure S5?

Line 139. "for" was wrote twic

Erban et al 2013 is missing in reference section.

Check that all cites in text are in references section and vice versa

Dear Editors and Reviewers,

Thank you for your comments on this paper. Below, we provide our responses, most of which occur in the form of additions or modifications of the existing manuscript. Reviewers comments are shown in italics, and our responses in normal text.

Response to Reviewer Comments

Reviewer #1 (Remarks to the Author):

Comment 1:

Dear Authors,

I carefully read the paper and the supplementary file provided with it as well as some relevant papers already published on the subject. The paper is about the links between aquifer over-pumping (mapped by detecting ground deformation using InSAR) and the release of arsenic from the aquitards (/clays layers) into the aquifer. It illustrates the important relation between aquifer sustainability and groundwater quality.

The methodology applied includes the use of a Machine Learning method that explores and reveals the link between different parameters, among which are considered compaction (/ground deformation), lithological data (/clay layer thickness), and arsenic concentration in groundwater. The paper is interesting and well written.

The phenomenological link between arsenic concentration and pumping has already been studied in the following paper (which btw, is missing in the reference list):

*Erban, L. E., Gorelick, S. M., Zebker, H. A., & Fendorf, S. (2013). Release of arsenic to deep groundwater in the Mekong Delta, Vietnam, linked to pumping-induced land subsidence. *Proceedings of the National Academy of Sciences*, 110(34), 13751-13756.*

Response and Changes: We have added it to the reference list.

Comment 2:

*I believe that Machine learning is interesting and will be increasingly used to explore how several observable parameters can be linked. It has the potential to trigger interesting phenomenological investigation. However, in this case of this paper, the phenomenology has already been explored. The only remaining novelty of the paper is the use of Machine learning to better transform proxies into the relevant information (here, arsenic concentration) and extrapolates point-based observation (training sites). As such, I don't believe that there is enough novelty within the paper to justify publication in *Nature Geoscience*.*

Response and Changes: Our manuscript presents the first statistical link between over-pumping and arsenic release, demonstrating quantitatively the extent to which over-pumping can degrade

groundwater quality. Furthermore, we establish this link in an area with a very different geochemical and geological environment than Vietnam, the study area of Erban et al. (2013). Arsenic arising from broadly-distributed groundwater pumping for irrigation has not been shown and illustrates a massive threat to groundwater quality. Indeed, the third reviewer felt that the mechanism we propose is, to date, ‘unproven’, and we provide evidence of the novelty of the method.

Reviewer #2 (Remarks to the Author):

The paper provides a quantitative machine learning model to estimate the arsenic concentrations using land subsidence in the San Joaquin Valley of California, because over pumping leads to dual consequences of land subsidence and increased extraction of pore water from arsenic-concentrated clay layers. The variables of the model have been evaluated using partial dependence plots. The methodology in the paper, if broadly applicable, will be valuable and efficient to water managements.

The paper is informative with interesting contents, but some objectives are not clearly described, in particular, need more explanations on Figures 3, 4, S1, and S3 to add credibility of this work. The paper can be considered to be published after major revisions according to the following comments.

Comment 1:

Missing Figure 2 in the main text.

Response and Changes: Figure 3 in the text should have been labeled Figure 2 and we have now corrected this oversight.

Comment 2: *Page 2 Line 40: Give full term and the abbreviation of As at its first appearance.*

Response and Changes: This has been corrected. See Comment 3 below for how the text now reads.

Comment 3: *Page 2 Line 41: Give abbreviation of “World Health Organization”, and use “WHO” consistently through the rest of the paper.*

Response and Changes: The text has been modified so it now reads:

“Approximately 10% of the wells tested within the last ten years have shown arsenic (As) concentrations above 10 ug/L, the level recommended as the maximum acceptable by the World Health Organization’s (WHO) recommended limit of 10 ug/L.”

Comment 4: *Page 3 Line 48: Typo of “Joaquin”.*

Response and Changes: This has been corrected so it reads: “Groundwater pumping in the San Joaquin Valley”

Comment 5: *Page 3 Line 60: In your statement, you chose two time frames (1986-1993 and 2007-2015) because they both span long droughts in the San Joaquin Valley. But from figure S5, the head declined from 1999 in the recent time frame. Why you pick up the dates from 2007 rather than 1999? And is there ground subsidence during 1999-2007?*

Response and Changes: The following was added to the existing text to address this concern: “Two models were developed, one that predicts recent (2007 to present) arsenic concentrations and a second that predicts historic (1986 to 1993) arsenic concentrations. These dates were chosen because they both span long droughts in the San Joaquin Valley, during which time appreciable groundwater decline and subsidence occurred (Figure S5). We did not include the mild-drought conditions from 1999 to 2005 because we had limited subsidence data over that period, and because we considered more intense droughts to produce a stronger signal for our analysis.”

Comment 6: *Page 4 Figure 1:*

Response and Changes: Figure 1 has been modified to address the reviewer’s comments. Here we respond to each comment individually.

The intervals for As concentration are 1, 3, 10, 32, 100 (~3-fold). Why you don’t use equal intervals like what you did for land subsidence? Due to high outliers? Provide the histogram and explain.

Response and Changes: We used a logarithmic scale to account for outliers and because arsenic concentrations tend to vary by several orders of magnitude. We modified the colorbar in the legend to improve clarity. We added a figure (S8) in the Supplementary Information to demonstrate this, and added the following text to the caption of Figure 1:

“Arsenic concentrations are shown with a logarithmic color bar because concentrations vary by several orders of magnitude (see Figure S8 for arsenic concentration histograms).”

The figure 1 in page 4 is a bit different from the full-page one you put separately: one has the inset and the other not, and one is rectangle and the other has been cropped into square. Put inset, and make sure the red box in the inset is in the same shape of the full-extent image.

Response and Changes: We have corrected this error so that both the in-page figure, and the one attached, are the same. We also verified that the outlined red box on the inset is in the correct location.

I can’t see clear spatial correlation between land subsidence and As concentration. The highest As concentration is located around 10 cm/yr land subsidence areas. Maybe it is easier to tell

with the interpolation results of As concentration and also contours like what you did for land subsidence.

Response and Changes: Indeed, there are many complex mechanisms that affect arsenic concentration, over-pumping being a major one. We added the following text to the methods section to emphasize this:

“Random forest model

Groundwater over-pumping, as indicated by subsidence, is one of many complex mechanisms that influence groundwater arsenic contamination. In order to account for additional mechanisms, as well as the interplay between various mechanisms, we employed the random forest model (Breiman et al., 2001). Random forest models can account for non-linear relationships between multiple variables and handle outliers well.~~We used the random forest model as described by Breiman et al. (2001).”~~

The areas in shadow show the InSAR coverage? Describe it in the caption.

Response and Changes: This has been added to the legend and caption.

What is the time frame(s) for the estimates of As concentration?

Response and Changes: 2007-2015. This has been added to the legend.

Comment 7. Page 4 Line 73: Provide the geologic vertical cross-section information about the confining clay layer in the text or in the supplementary. Or add reference.

Response and Changes: The caption to Figure 1 now reads:

“Figure 1: Comparison of subsidence, arsenic concentration, and clay extent within the San Joaquin Valley, CA. Arsenic concentrations increase towards the center of the valley where subsidence is more extensive and a confining clay layer (known as the Corcoran clay) is present (Page et al., 1986). The grayed-out region is the area where recent subsidence data (obtained from InSAR) were processed. Arsenic concentrations vary by orders of magnitude and are thus shown with a logarithmic color bar (see Figure S8 for arsenic concentration histograms).”

Comment 8: Page 6 Figure 6:

The land subsidence is between 0 and 25 cm/yr during 2007-2011. Why you use a narrow range of 0 to ~10 cm/yr in (d)?

Response and Changes: There were very few wells (<5%) with arsenic concentration data that had greater than 10 cm of subsidence, so the model is not well constrained for levels exceeding 10 cm. To explain our decision for using the 0 to 10 cm range, we added the following section to the methods on the partial dependence plots:

“Random forest models are limited in their predictions to observed data. For this reason, in the partial dependence plots of the main article (Figures 2 and 3), we restricted the independent variable at the 5th and 95th percentile of each predictor variable. With those limits, 90% of the available data are displayed in the partial dependence plots. This reduces the likelihood of presenting results which are unconstrained by data. In the supplementary information section, we presented all of the data, but displayed dashed lines indicating the 5th and 95th percentiles.”

Here in Figure 6 you show the partial dependence plots for primary variables used in the model predicting arsenic concentration from 2007-2015 (Figure S1). There are some differences between the results of 2007-2015 and 1986-1993 (Figure S3), such as partial dependence plots for the thickness of Corcoran clay and Mn concentration. Please explain the possible reasons.

Response and Changes: “The historic arsenic concentration model shows a weaker, but still positive, correlation between Corcoran clay thickness and arsenic concentration, as well as manganese concentration and arsenic concentration. Due to the dynamic nature of groundwater chemistry, the relative importance of different mechanisms for arsenic contamination change over time, highlighting the need for regular calibration with up to date datasets.”

I suppose you draw (e) according to the “sub86to93” in Figures S1 and S3, and take the one in Figure S1 as the reference? Why the x axis is 0 to 2 in the supporting figures, but 0-20 cm/yr in (e)?

Response and Changes: The different values are a result of different units. Since this is likely to confuse other readers, we changed the units to cm/yr in all subsidence figures, and labeled the units in the supporting figures.

Comment 9: *Page 7 Line 115: Explain how you infer “recent subsidence has a great effect on recent arsenic levels” from Figure 3d.*

Response and Changes: Perhaps it was not clear that Figure 3d (now labeled Figure 2d to correct a numbering error) is showing recent arsenic levels. As recent subsidence increases, so does the recent arsenic level. This was one of the most important mechanisms as determined by the random forest model, which inferred no prior relationships. To clarify that figure 3d is referring to recent arsenic levels, we added the following to the end of the caption:

“All blue lines are derived from models predicting recent (2007-2015) arsenic concentrations, while the red line is derived from models predicting historic (1986-1993) arsenic concentrations.”

Comment 10: *Page 7 Lines 125-129: Rearrange the sentences. “As the rate of subsidence increase, inelastic deformation of clay beds also increases significantly”. This sentence is not rigorous.*

Response and Changes: Since deformation of clay beds causes subsidence, and not the other way around, the sentences were rearranged per the reviewer’s suggestion so the text now reads:

“As inelastic deformation of clay beds begins, the rate of subsidence increases; ~~inelastic deformation of clay beds also increases~~ significantly (Smith et al., 2017).”

Comment 11: *Page 8 Figure 4: Be consistent about the time period of land subsidence, 2007-2010 or 2007-2011 (shown in figure 1).*

Response and Changes: We modified Figure 1 to show subsidence from 2007-2010 to correct this error.

Comment 12: *Page 11 Lines 195: Add sensor name and time period.*

We added this information to the text, it now reads:

“We acquired 68 SAR scenes from the ALOS PALSAR sensor, covering the time period from 2007 to 2010, over four frames for InSAR processing from the Alaska Satellite Facility (<https://www.asf.alaska.edu/>).”

Comment 13: *Page 11 Line 203: Is there seasonality in the time series? If the seasonality need to be addressed in your model?*

Response and Changes: We addressed this question with the following addition:

“Because the long-term signal dominated the total signal, seasonal fluctuations are muted over a multi-year time period. Thus, because of the temporal resolution of our arsenic measurements, we only considered the long-term signal.”

Comment 14: *Supporting material: Page 11 Figure S7: Why are there voids in the estimates?*

Response and Changes: This question was addressed by adding the following to the caption:

“Estimated historic groundwater flow per m² cross-sectional area, in m/day. The color scale is logarithmic. This dataset was created by multiplying the land surface slope by the estimated hydraulic conductivity from Faunt et al. (2009), where both datasets exist.”

Comment 15: *Supporting material: Pages 4 and 7: What is the meaning of “sub76”? Do you mean “subsidence from 1962 to 1976” (Page 10 Line 163)?*

Response and Changes: The label was left out of Table S1, and has been added:

Variable name	Variable description	Source
---------------	----------------------	--------

Reviewer #3 (Remarks to the Author):

Authors present a very interesting approach which have high practical application potential. Nevertheless, some points have to be clarified and probed in order to increase and strengthen the facts and evidences in which they support their proposal.

Authors based their proposal in the unproved fact that arsenic is being expelled from clay strata due to the subsidence process, and thus subsidence maps can be used to detect where arsenic concentrations will be have place.

Comment 1: *To prove this, authors have to present evidences (or specific references for the study site) showing that the strata clay (or the water which is filling clay pores) have a high content of arsenic.*

Authors stated in line 75 that "The relationship between subsidence and arsenic concentration is linked to clay strata". Nevertheless, figure 1 shows that some of the arsenic highest concentration locations (roughly 20 %) are located outside the limits of the clay formation. How authors explain that? Is there clay samples lab analysis to affirm that arsenic comes from clay?

Response and Changes: The main concern of the reviewer is that we provide evidence that there is higher arsenic content in clay strata within our study area. We address this comment by providing specific laboratory evidence for arsenic in clays in our study area (Gao et al., 2004). We also cite Fujie and Swain (1995) who found that wells below 60 m tend to have reduced arsenic, providing the mechanism for arsenic dissolution in groundwater. The revised paragraph is shown below:

“The relationship between subsidence and arsenic concentration is linked to clay strata (Figure 1). Fendorf et al. (2010) noted that clay particles tend to become enriched in arsenic due to their reactivity and high surface area relative to sand- or silt-sized particles. Arsenic has been transported to the San Joaquin Valley from the Sierra Nevada and coastal mountain ranges by rivers, which cut through arsenic-bearing formations, for millions of years (Fujie and Swain, 1995). Clays at or near the surface at the time of deposition are the primary host of transported arsenic, which has been shown to adsorb on clay surfaces in significant amounts in the San Joaquin Valley (Gao et al., 2004), in a process similar to what occurs in other sedimentary basins, including those throughout Asia (Fendorf et al., 2010). As the clays are buried over geologic time, their restricted oxygen supply results in reduction of the arsenic at depths greater than 60 m (Fujie and Swain, 1995), resulting in dissolution of the arsenic within the clay pore-water, and clay formations restrict oxygen supply, promoting arsenic reductive dissolution into the pore water (Erban et al., 2013) (Erban et al., 2013). This mechanism for arsenic concentration in clays is supported by Ayotte et al. (2017), who found a positive relationship between aquifer clay content and arsenic concentration in the San Joaquin Valley.”

The reviewers also note that there are some high-arsenic wells (roughly 20%) outside of the Corcoran clay. We did not assert that clay content is the *only* source of arsenic contamination, but find it striking that ~80% of high-arsenic wells do occur within the Corcoran clay.

Comment 2: *Other factors can be cause of anomalous arsenic concentrations, and authors have to discuss them. For example:*

The use of pesticides surely has been extensive during long time in the area, as the valley is a very productive crop area. Then pesticides infiltrations could be linked to the current arsenic concentrations.

Deep ground water has having enough time to interact with rocks (bedrock or coarse sediments), consequently it could result in natural pollution by substances as arsenic.

How authors discarded those potential arsenic sources?

Response and Changes: To address this comment, we added the following to our **Methods** section, under *Data acquisition and processing*

We chose not to include pesticide application as a predictor of groundwater arsenic contamination. Arsenic-bearing pesticides are largely restricted to lead arsenicals that were used predominantly on cotton and specific tree fruits (largely apples) but have been banned since the 1970's within the U.S. Within the aerated surface soils, arsenic exists as arsenate (Gao et al., 2004) and would have limited mobility (see Fendorf et al., 2010). Thus, downward migration of surface applied arsenic is unlikely to contribute to groundwater contamination. Further, our analysis shows a correlation of groundwater arsenic with recent (last 10 years) subsidence and no correlation with older periods of subsidence. Our time-series therefore further removes the possibility of arsenic pesticides impacting groundwater.

Comment 3, Other minor observations to improve are:

Sections are not clearly delimited as a typical article (i.e. introductions, methodology, results, discussion of results, conclusions, etc.).

Response and Changes: We leave this decision to the Editors. We see the article as providing a coherent message, but are happy to change the format if the Editors feel it is necessary.

Lines 48-50. Those data are not very current. I suggest leave it as antecedent but include updated information and their references... I suppose that this information exists since this valley is one of the most studied in the world.

Response and Changes: To clarify this statement, we modified the text as follows:

“Groundwater pumping in the San Joaquin Valley has caused declines of ~60 m in groundwater levels over the past century, leading to subsidence as high as ~~~910 m~~ over this time period from 1925 to 1970, a rate of 20 cm/year (Poland et al., 1975). Rates of subsidence as high as 25

cm/year have been observed in more recent droughts spanning from 2007-2010 (Smith et al., 2017).”

Lines 40-51. Terzaghi, 1925 deal with soil consolidation due to an overburden on top of soil stratum, and not about subsidence due to groundwater withdrawal. I suggest change this reference by one dealing with the phenomenology of subsidence due to groundwater extraction.

Response and Changes: The theory of consolidation developed by Terzaghi (1925) states that $\sigma_e = \sigma_T - P_p$, where σ_e is the effective stress, σ_T is the total stress (overburden) and P_p is the pore pressure. Extracting groundwater does not affect the overburden, but does reduce the pore pressure, which reduces the effective stress. We clarified this in the text as follows:

“Subsidence due to groundwater pumping is caused by a pore pressure drop in aquifer materials, which increases the effective stress, σ_e , defined as $\sigma_e = \sigma_T - P_p$, where σ_T is the total stress, and P_p is the pore pressure (Terzaghi, 1925). Increasing the effective stress results in ~~their~~ aquifer compaction (Terzaghi, 1925).”

Lines 51-53. Authors pointed out that "The majority of compaction and linked subsidence of the overlying ground surface is caused by drainage of clays due to their weaker geomechanical properties."

The term "weaker geomechanical properties" could be confusing. I suggest that it be specified which geomechanical properties.

Response and Changes: We clarified this with the following text:

“The majority of compaction and linked subsidence of the overlying ground surface is caused by drainage of clays due to their weaker geomechanical properties, as indicated by their higher skeletal specific storage.”

Line 63. Figure 1 instead of Figure S5?

Response and Changes: The reviewer refers to this line: “These dates were chosen because they both span long droughts in the San Joaquin Valley, during which time appreciable groundwater decline and subsidence occurred (Figure S5)”

Figure S5 shows a time series of head decline, which can be viewed to understand the times with the most significant droughts. We feel this is the appropriate figure to reference in the text.

Line 139. “for” was wrote twice

Response and Changes: This error has been fixed.

Erban et al 2013 is missing in reference section.

Response and Changes: This reference has been added.

Check that all cites in text are in references section and vice versa

Response and Changes: We have removed references from the manuscript that are not cited in the text and added references that were cited but missing from the references list.

REVIEWERS' COMMENTS:

Reviewer #2 (Remarks to the Author):

The authors replied to the comments in a satisfying way. However, there are still some issues that should be improved further.

L59. Provide information about the sedimentary structure of aquifers in your study area, such as the depth, sequence, and thickness of the sands, gravels, silts, and clays. The authors need to give the readers a picture of the structure before mentioning "sand- or silt-sized particles" in L89 and "sands and gravels" in L103. The statement in L89 that the clay particles have "high surface area relative to sand- or silt-sized particles" is confusing.

L89. Explain the reactivity.

L188. "2007 to present" should be "2007 to 2015".

Supporting information. "Figures S1 to S7" should be "Figures S1 to S8".

Dear Editors and Reviewers,

Thank you for your comments on this paper. Below, we provide our responses, most of which occur in the form of additions or modifications of the existing manuscript. Reviewers comments are shown in italics, and our responses in normal text.

Response to Reviewer Comments

Reviewer #2 (Remarks to the Author):

The authors replied to the comments in a satisfying way. However, there are still some issues that should be improved further.

Comment 1:

L59. Provide information about the sedimentary structure of aquifers in your study area, such as the depth, sequence, and thickness of the sands, gravels, silts, and clays. The authors need to give the readers a picture of the structure before mentioning “sand- or silt-sized particles” in L89 and “sands and gravels” in L103.

Response and changes: We added a description of the sedimentary structure, as shown below:

Thus, the high subsidence levels provide a measure of over-pumping in aquifers having substantial clay content, such as the aquifers of the San Joaquin Valley. In this region, the upper 500 m, which is the greatest depth typically drilled for groundwater pumping, consists of alternating layers of sand, gravel, and clay. Generally speaking, the aquifer system is divided into an upper aquifer, a thick clay confining unit known as the Corcoran clay, and a lower aquifer. The upper and lower aquifers contain sands and gravels, as well as numerous thin clay layers⁸.

Comment 2:

The statement in L89 that the clay particles have “high surface area relative to sand- or silt-sized particles” is confusing.

Response and changes: We have clarified the text as shown below:

The relationship between subsidence and arsenic concentration is linked to clay strata (Figure 1). Fendorf et al. (2010) It has been noted that clay particles become enriched in arsenic due to their reactivity and high surface area to volume ratio relative to sand ~~or silt~~-sized particles⁴.

L89. Explain the reactivity.

Response: At the terminal faces of clay minerals, ionizable functional groups give rise to electrostatic charge, to which counterions in the solution bind through physical forces. Further, the coordinatively undersaturated hydroxyl groups on the mineral surface form ionic or covalent bonds with compatible ions. In the case of arsenic, arsenate has a propensity to form chemical bonds (sometimes referred to as inner-sphere complexes) with clay mineral surfaces. Given the general knowledge of clay mineral reactivity, no changes were made.

Comment 3:

L188. “2007 to present” should be “2007 to 2015”.

Response: This change has been made

Comment 4:

Supporting information. “Figures S1 to S7” should be “Figures S1 to S8”.

Response and changes: We changed this to ‘Supplementary Figures 1 to 8’ to match the Nature style guide.